# Operational Details of the Five Domains Model and Its Key Applications to the Assessment and Management of Animal Welfare

**DOI:** 10.3390/ani7080060

**Published:** 2017-08-09

**Authors:** David J. Mellor

**Affiliations:** Animal Welfare Science and Bioethics Centre, Institute of Veterinary, Animal and Biomedical Sciences, Massey University, Palmerston North 4442, New Zealand; d.j.mellor@massey.ac.nz; Tel.: +64-6-356-9099 (ext. 84024)

**Keywords:** affects, five domains model, model applications, situation-related factors, survival-critical factors, welfare assessment, welfare management

## Abstract

**Simple Summary:**

The Five Domains Model is a focusing device to facilitate systematic, structured, comprehensive and coherent assessment of animal welfare; it is not a definition of animal welfare, nor is it intended to be an accurate representation of body structure and function. The purpose of each of the five domains is to draw attention to areas that are relevant to both animal welfare assessment and management. This paper begins by briefly describing the major features of the Model and the operational interactions between the five domains, and then it details seven interacting applications of the Model. These underlie its utility and increasing application to welfare assessment and management in diverse animal use sectors.

**Abstract:**

In accord with contemporary animal welfare science understanding, the Five Domains Model has a significant focus on subjective experiences, known as affects, which collectively contribute to an animal’s overall welfare state. Operationally, the focus of the Model is on the presence or absence of various internal physical/functional states and external circumstances that give rise to welfare-relevant negative and/or positive mental experiences, i.e., affects. The internal states and external circumstances of animals are evaluated systematically by referring to each of the first four domains of the Model, designated “Nutrition”, “Environment”, “Health” and “Behaviour”. Then affects, considered carefully and cautiously to be generated by factors in these domains, are accumulated into the fifth domain, designated “Mental State”. The scientific foundations of this operational procedure, published in detail elsewhere, are described briefly here, and then seven key ways the Model may be applied to the assessment and management of animal welfare are considered. These applications have the following beneficial objectives—they (1) specify key general foci for animal welfare management; (2) highlight the foundations of specific welfare management objectives; (3) identify previously unrecognised features of poor and good welfare; (4) enable monitoring of responses to specific welfare-focused remedial interventions and/or maintenance activities; (5) facilitate qualitative grading of particular features of welfare compromise and/or enhancement; (6) enable both prospective and retrospective animal welfare assessments to be conducted; and, (7) provide adjunct information to support consideration of quality of life evaluations in the context of end-of-life decisions. However, also noted is the importance of not overstating what utilisation of the Model can achieve.

## 1. Introduction

Fresh conceptual frameworks are usually developed to correct perceived errors or inadequacies in current ideas and thus owe a debt to the earlier ways of thinking from which they evolved. In animal welfare science, the 1994 formulation of the Five Domains Model [1] had its origin in the highly influential Five Freedoms paradigm [2,3,4,5,6]. However, cursory consideration of this original linkage has led to an erroneous view that the Model is merely a substitute for the Five Freedoms [7,8], whereas the Five Provisions/Welfare Aims paradigm has recently been formulated for that specific purpose [6].

The influence of the Five Freedoms paradigm over the last two decades was likely due to four key factors [5,6,7], namely that it (1) scoped the wider dimensions of animal welfare, including subjective experiences, health status and behaviour; (2) specified areas of welfare concern in terms of particular negative experiences (thirst, hunger, fear, distress, discomfort, pain) and states (malnutrition, injury, disease, behavioural expression); (3) defined five particular targets for welfare improvement in terms of “Freedoms”; and (4) presented practical advice on how these targets might be achieved by outlining particular “Provisions” aligned with each Freedom. Thus, the Five Freedoms paradigm became so integrated into the understanding of animal welfare and its management that it, at least implicitly, assumed a status akin to a definition [3,4]. Broom [9] defined animal welfare more explicitly as “the state of an animal as regards its attempts to cope with its environment”, adding that, “welfare is a wide term that embraces both the physical and mental well-being of the animal”.

The Five Domains Model was designed specifically to facilitate structured, systematic, comprehensive and coherent animal welfare assessments, with a focus initially on welfare compromise [1,10,11] and then on both compromise and enhancement [12]. The Model was never intended to have an implicit or explicit role as a definition of animal welfare. Thus, its regular updates [10,11,12,13] using the latest validated knowledge (e.g., [5,14,15,16,17,18,19,20,21,22]) were designed to sustain the breadth, depth and currency of Model-based welfare assessments in order to improve animal welfare management, and not to give the Model any standing as a definition of animal welfare. Moreover, the principal architect of the Model (the present author), in acknowledging that little consensus has emerged among scientists regarding definitions of animal welfare (see [11,23,24,25]), has long preferred to characterise animal welfare in terms of its currently accepted major attributes (e.g., [5,11]). This avoids the potential inflexibility and defensiveness that definitions sometimes attract, allows well-accepted notions to be included and, as ideas change, for related features of the characterisation to be revised appropriately or discarded [5].

In view of these observations, and growing interest in utilising the Five Domains Model in various animal use sectors [1,10,11,26,27,28,29,30,31,32,33,34,35,36,37], it would be helpful to clarify what can and cannot be accomplished by knowledgeable application of the Model to the assessment and management of animal welfare. Accordingly, unlike previous publications where the focus was on the scientific foundations of the Model [5,11,12,29], the approach adopted here is to emphasise key operational features of the Model. To this end the present paper begins with a brief outline of those features of the Model that support its use for welfare assessment. It continues by providing helpful examples of states, situations, affects and interactions between domains to illustrate key elements of how the Model operates. Then seven key applications of the Model to welfare assessment and management are enunciated, and they are followed by conclusions.

At the outset, note that consideration of what animals experience subjectively, i.e., their affects, has a key role in contemporary animal welfare science thinking [5,12,33,38], the affects of welfare significance being those that are consciously experienced as unpleasant or pleasant rather than as hedonically neutral [8,14,17,18,24,25,39,40]. The occurrence of the affects animals may experience is inferred from the presence of internal states and/or external conditions responsible for generating them [5,12]. Accordingly, every evaluation of an animal’s general welfare status, or specific features of it, is hypothetical to the extent that it involves making such inferences [38]. However, those inferences derive credibility from validated knowledge of the underlying systems physiology, neurophysiology and affective neuroscience, as also from the caution exercised when inferring the presence of particular affects (e.g., [15,19,22,38,41,42]). Thus, the process involves cautiously exercising scientifically informed best judgement.

## 2. Major Features of the Five Domains Model

The major features of the Model depicted in Figure 1 and explained briefly here have been outlined in detail in a series of fully referenced review articles [1,10,11,13,28,43]. These are the primary sources for the following brief account; other sources are noted below. A fundamental aspect of the Model’s use is that all interpretations should be credibly supported by current scientific knowledge.

### 2.1. General Overview of the Model

The Model is not intended to be an accurate representation of body structure and function. Rather, it is a focusing device designed to facilitate assessment of animal welfare in a systematic, structured, comprehensive and coherent manner. The purpose of each of the five domains is to draw attention to areas that are relevant to welfare assessments. The Model therefore facilitates identification of internal physical/functional states and external circumstances that give rise to negative and/or positive subjective mental experiences (affects) that have animal welfare significance. As the body functions as a dynamically integrated whole entity, the specific body functions or states, external circumstances and related affective experiences identified via the Model inevitably interact. Accordingly, there may be overlap between factors considered within different domains. However, awareness of the potential for this avoids problems when using the Model, as illustrated below (see Section 2.3 and Section 3 ).

### 2.2. The Domains and Their Role

Specifically, the Model focuses attention on welfare-significant *internal states* via Domains 1 to 3, which are labelled “Nutrition”, “Environment” and “Health”, and on welfare-significant *external circumstances* via Domain 4, which is labelled “Behaviour”. Once the internal states and external circumstances have been identified, any associated affective experiences, inferred cautiously, are accumulated into Domain 5, which is labelled “Mental State”. The indices of the *internal states* are mainly anatomical, biochemical, physiological and clinical in character, but also include behaviour, whereas the indices relevant to consideration of the impacts of *external circumstances* are mostly behavioural, but they may be supported by some of the functional/clinical indices just mentioned.

Regarding Domains 1 to 3, “Nutrition”, “Environment” and “Health”, the affects generated by particular internal states are understood to be genetically pre-programmed drives that impel animals to engage in behaviours that help to secure their survival. These are designated survival-critical behaviours and affects. Each such affect motivates a specific behaviour (e.g., breathlessness drives heightened respiratory activity, thirst drives water seeking and drinking, pain motivates withdrawal from or avoidance of injurious events), and their undoubted negativity is essential to create a sense of urgency or a compulsion to respond.

The *valence*, i.e., the emotional character, of these affects is confined to the *negative-to-neutral range* (Figure 2). Assigned to Domain 5, “Mental state”, these affects are now considered to include breathlessness, thirst, hunger, pain, nausea, dizziness, debility, weakness and sickness. Nevertheless, behaviours motivated by some of these negative affects may also provide opportunities for animals to have positive experiences; for example, thirst-motivated water intake may give rise to the wetting and quenching pleasures of drinking, and hunger-motived foraging or hunting may lead to the consumption of foods having pleasant tastes, smells, textures and variety (Figure 1).

The role of Domain 4, labelled “Behaviour”, is to focus attention on animals’ likely perceptions of their external circumstance and the affective experiences that may be associated with those perceptions. To emphasise this, Domain 4 was assigned the role of dealing with “*situation-related factors*”. Behaviour, incorporating appearance, demeanour, activity/inactivity and vocalisation/silence, evaluated in the context of the animals’ physical, biotic and social environment, guides making cautious inferences about the affects they are likely to experience, accumulated into Domain 5, designated “Mental state”.

Close confinement and isolation of social animals in threatening and/or barren environments may lead to experiences that include various combinations of anxiety, fear, panic, frustration, anger, helplessness, loneliness, boredom and depression (Figure 1 and Figure 2). In domesticated circumstances, these external conditions are often imposed by persons in charge of the animals, and, if so, are amenable to corrective measures being taken by them.

Keeping social animals with congenial others in spacious, stimulus-rich and safe environments provides them with opportunities to engage in behaviours they may find rewarding, in other words, it provides opportunities for them to experience “positive affective engagement” [20]. In general terms, the associated positive affects are considered likely to include various forms of comfort, pleasure, interest, confidence and a sense of being in control, and, more specifically, may include the following feelings: being energised, engaged, affectionately sociable; rewarded maternally, paternally or as a group when caring for young; and being nurtured, secure or protected, excitedly joyful, and/or sexually gratified (Figure 1) [5,21,22].

Providing improved or enriched external circumstances enables animals to exercise agency with potentially positive affective outcomes; i.e., they have greater opportunities to engage in voluntary, self-generated and goal-directed behaviours that they may find rewarding [12,44,45]. Such improvements or enrichments may focus on very few activities; some may be intermittently applied to retain novelty and interest over time; they may involve a range of features thereby providing a wider choice of pleasurable opportunities at any one time; and/or the animals may be given access to stimulus-rich natural environments which they demonstrably find engaging (e.g., [34,46]). Thus, improving or enriching impoverished external circumstances can lead to some situation-related negative affects being replaced by positive ones [20,21,22]. The *valence* of the full spectrum of such situation-related affects encompasses the *negative-through-neutral-to-positive range* (Figure 2) [5,20].

### 2.3. Interactions between Negative Survival-Critical and Positive Situation-Related Affects

There are potential interactions between the negative affects generated by physical/functional imbalances or disruptions, captured by Domains 1 to 3, and the motivation of animals to engage in rewarding behaviour, captured by Domain 4 [12,37]. When the intensity of such negative affects is significant (Domain 5), animals usually do not engage in rewarding behaviours even when opportunities to do so are available (Figure 2) [5,12,37].

### 2.4. Neuroscience Support for the Identification of Particular Affects

An increasing understanding of the brain processing that underlies aversive and rewarding experiences and their manifestation as specific affects (e.g., [16,47]) provides support for cautiously inferring that specific internal states and/or expressed behaviours are suggestive of animals experiencing particular negative or positive affective states. Thus, a considerable amount of evidence from what has come to be known as affective neuroscience now supports such inferences, made cautiously, and thereby successfully challenges accusations of anthropomorphism (see [16]).

As the focus of Model-based welfare assessments is on specific affects, or groups of affects, and their sources, it is important to consider how well supported are inferences about the presence of each affect. Confidence in such inferences depends on how well-described the underlying affective neuroscience is, the specificity of any physical/functional indices and/or the distinctiveness of indicative behaviours, all evaluated in the context of the animal’s physical, biotic and social environment.

Regarding the *negative survival-critical affects*, the underlying neuroscience knowledge, physical/functional indices and behaviours are well demonstrated and allow breathlessness, thirst, hunger, pain and sickness to be identified (e.g., [15,19,20,41,42,48,49]). However, it is not as easy to distinguish between nausea, dizziness, debility, weakness and sickness unless the specific circumstances of the animal and/or specific functional indices provide sufficient justification to identify a particular affect [15]. If not, two or more of these affects might be combined, for example, nausea and dizziness or debility, weakness and sickness, allowing for less specific, but still informative consideration.

In the *situation-related category,* there are good neuroscience bases for using indicative behaviours to cautiously distinguish between the *negative affects* of anxiety, fear, panic, depression, frustration and anger observed in animals in relation to their particular circumstances (e.g., [15,16,47,50,51,52,53,54], whereas behavioural indices may not enable helplessness, loneliness and/or boredom to be distinguished as easily. Accordingly, identifying any of the latter group of affects should be done with even greater caution. This caveat is not intended to cast doubt on their existence; it is just to note that they may be difficult to identify or distinguish from each other behaviourally.

Regarding *positive affects* in the *situation-related* category, affective neuroscience observations underpin interpretation of particular behaviours as indicating experiences of “positive affective engagement” [5]. More specifically, the neuroscience of reward seeking and generation of positive affects supports the interpretation that animals will likely have pleasurable experiences when engaged in the following behaviours [21]: positively motivated environmental exploration and food acquisition activities; bonding and bond affirmation; maternal, paternal or group care of young; play behaviour; and sexual activity (e.g., [16,17,55,56,57,58,59,60,61]).

## 3. Examples of States, Situations, Affects and Domain Interactions Relevant to Utilisation of the Model

In addition to showing the structure of the Model, Figure 1 also provides numerous examples within each domain. These examples are indicative, not definitive or comprehensive. Each example should be assessed by reference to the animals’ species-specific behaviour, biology and ecology considered in relation to their specific physical, biotic and social environment. The examples in Figure 1 may therefore be retained, deleted or amended, and/or others added as deemed appropriate for each species (e.g., [37]).

Affects considered to be associated with *physical/functional states* that may be normal, disrupted or out of balance, and/or affects associated with *behaviours* that may indicate impeded, unimpeded or enhanced exercise of agency (see Figure 1), should be included only when there is credible scientific support for their alignment with those states or behaviours. In the absence of such support, inferences about such affects should be avoided. For example, this might rule out confident speculation about the precise affective experiences that may be generated by sensory inputs that have no known human equivalent, such as those associated with echolocation and hearing in the ultrasonic range. Note, it is not suggested that there is no affective impact of such sensory inputs; rather that understanding the precise nature of the affects remains problematic.

The dynamically integrated functionality of the whole body (already noted) means that some factors considered in welfare assessments will interact across domains. This is inevitable. Three examples of *survival-critical factors* illustrate this:
Feeding levels (Domain 1) that otherwise minimise hunger (Domain 5) would be inadequate when animals need to forage over long distances on sparse pasture and fail to meet the additional energy intakes required to support that exercise (Domain 4);Cold ambient conditions that increase energy demands for heat production (Domain 2) in animals otherwise fed at adequate levels (Domain 1) would likely add chilling discomfort to elevated intensities of hunger (Domain 5);Respiratory discomfort, for example breathlessness (Domain 5), may be due to atmospheric pollutants (e.g., ammonia) (Domain 2), lung pathology (e.g., pneumonia) (Domain 3) or sustained exercise at the upper limit of athletic capacity (e.g., escape from predators; racing at near maximum speed) (Domain 4), where the precise aetiology in each case would differ.

Although consideration of the welfare impacts of *situation-related* behaviours has been assigned primarily to Domain 4 of the Model, *survival-critical* behaviours aligned with situations relevant to Domains 1 to 3 are also important, as illustrated by the following three examples [12]:
Water-seeking and drinking motivated by thirst and foraging/hunting motivated by hunger are behaviours relevant to Domain 1;Seeking out warm or cool environmental locations and/or adopting appropriate thermoregulatory postures in them are behaviours relevant to Domain 2;Withdrawal from and/or avoidance of injurious stimuli that cause pain are behaviours of relevance to Domain 3.

The Model design facilitates identification of such cross-domain interactions, so that users should remain flexible in their allocation of specific factors to each domain, guided by common sense and scientific knowledge. Arguments based on rigid allocation of a factor to a particular domain are avoided by ensuring that, in each situation, each factor and its aligned specific affects are considered only once. Recall that the Five Domains Model is a facilitatory device. Its flexibility is a major strength. It is not a rigid construct that must be adhered to dogmatically.

## 4. Key Model Applications

The primary role of the Model is identification of key internal and external factors that contribute to the generation of specific negative and positive affects of welfare significance to provide a basis for systematic, structured, comprehensive and coherent animal welfare assessment. Aligned with this role are seven overlapping major applications of the Model that support effective animal welfare management.

### 4.1. Application 1: The Model Specifies Key General Foci for Animal Welfare Management

The Model directs attention towards general areas of welfare concern related to negative affects and/or towards opportunities for welfare enhancement related to positive affects. It is by highlighting the internal and/or external origins of these affects that the Model provides guidance regarding the general targets for animal welfare management activity. In practical terms, meeting these welfare management targets involves husbandry and veterinary activities, the availability of resources and the suitability of facilities, collectively known as the Provisions [2]. These Provisions were updated recently [6], such that the first four align with the physical/functional domains of the 2015 Model (Figure 1) [12] and reflect the four principles of the European Welfare Quality (WQ^®^) system [62,63]. They are designated “good nutrition”, “good environment”, “good health” and “appropriate behaviour”. These Provisions represent the key general foci of animal welfare management. The fifth Provision, designated “positive mental experiences” [6], aligns with Domain 5 of the Model (Figure 1) and emphasises the promotion of positive welfare states. Accordingly, the *motivation* for taking required welfare-focused actions and the general direction of these actions are based on understanding the origins of specific affects (see Application 2). However, their *practical management* is achieved through knowledgeable interventions focused on the first four updated Provisions [5,6]. Thus, it is not necessary to be able to measure affects directly to manage them practically [5].

### 4.2. Application 2: Model Use Highlights the Foundations of Specific Welfare Management Objectives

*Survival-critical negative affects*, the negativity of which is essential to create a sense of urgency or a compulsion for animals to engage in behaviours directed at acquiring life-sustaining resources (e.g., oxygen, water, food) and/or avoiding or minimising potentially fatal threats (e.g., injury, food poisoning, infection), can never be eliminated [5,25]. Thus, the specific welfare objective related to these affects (Figure 1 and Figure 2) is to apply animal care strategies that reduce their intensities and occurrence to low/tolerable levels that nevertheless still motivate the essential behaviours when they are needed [12,20]. Minimising these affects does not, in and of itself, lead to a positive net welfare balance, but it may reduce the inhibitory effects they may have on animals’ motivation to engage in rewarding behaviours (see below) [12,20,37]. The *valence* of these *survival-critical affects* resides in the *negative-to-neutral range* (Figure 2).

*Situation-related negative affects* relate to animals’ perception of their external circumstances and are linked especially to isolation, low stimulation, inadequate space, threat, and/or to restrictions on the exercise of agency [21]. Their *valence* resides in the *negative-to-neutral range* (Figure 2). The specific welfare objective related to these negative affects is to *replace* them with positive affects by improving or enriching the animals’ environment in ways that provide greater opportunities for them to engage in behaviours they find rewarding [12,21,22].

*Situation-related positive affects* relate to animals’ perception of external circumstances that enable them to experience various forms of comfort, pleasure, interest and confidence, as well as a sense of being in control through the exercise of agency and the utilisation of opportunities to engage in rewarding behaviours [20,21]. The *valence* of these affects resides in the *neutral-to-positive range* (Figure 2).

As already noted (Section 2.3), *survival-critical negative affects* and *situation-related positive affects* may interact. When the intensity of one or more of the former negative affects is above tolerable levels, this may demotivate animals from utilising existing opportunities to engage in behaviours (e.g., exploring, foraging/hunting, affirming bonds, playing) that would likely be accompanied by positive affective experiences (Figure 2) [12,37].

In summary, the key objectives of animal welfare management that determine the specific practical applications of the Provisions are both to minimise negative survival-critical and situation-related affective experiences, and to provide opportunities for animals to have positive situation-related affective experiences [12,20,21,22]. More specifically, keeping survival-critical negative affects at low/tolerable levels that nevertheless still motivate the required behaviours is important to reduce the intensity of those particular forms of welfare compromise in their own right. However, this is doubly important because it also helps to minimise the potential inhibitory impacts that these negative affects may have on animals’ motivation to engage in rewarding behaviours. Finally, providing animals with opportunities to engage in such rewarding behaviours is important to enable situation-related negative affects to be replaced with positive ones (Section 2.2 and Section 2.4), because this is a major way the overall welfare state of animals can be improved [12,20].

### 4.3. Application 3: Model Use Helps to Identify Previously Unrecognised Features of Poor and Good Welfare

In the past, and still today, it is common to describe exceptionally unpleasant experiences using the catch-all term “suffering”, where “suffering” has been taken to include, for example, “pain”, “mental cruelty”, “discomfort” and “distress” [2,11,12,64,65]. Yet these descriptors, including “pain”, which in fact has many manifestations [15], are as imprecise as the term “suffering” when considering what specific affects may be involved (e.g., breathlessness, thirst, hunger, nausea, sickness, fear, panic), and in that sense they are all generic. It is more informative when deciding what remedial actions should be taken to have a capacity to identify what specific affects are contributing to the negative states under consideration. Accordingly, the 2009 version of the Model was formulated to draw attention to a wider range of negative affective experiences, which, when at intensities greater than tolerable levels, may represent specific forms of “suffering” or “distress” [11,29]. As already noted, the most recent version of the Model [12] also includes a range of positive experiences related to animals’ comfort, pleasure, interest and confidence, as well as to their sense of being in control through the exercise of agency and the utilisation of opportunities to engage in rewarding behaviours [20,21].

Thus, the numerous examples of internal states and external circumstances, and their potentially aligned affective experiences now incorporated in the Model, draw attention to wide ranges of specific negative and/or positive welfare-relevant experiences (Figure 1). This has three beneficial outcomes which support Applications 1 and 2, namely: (1) the Model alerts managers and animal care staff to a wide range of welfare-relevant states and situations and the aligned experiences animals may have which they would not previously have considered; (2) when corrective action is required, the Model enables remedies to be more specifically focused, thereby improving the likelihood of expeditious beneficial outcomes; and (3) the Model potentially engenders greater empathy towards animals, thereby directing more attention towards their care and not merely to their routine maintenance.

### 4.4. Application 4: Model Use Enables the Monitoring of Responses to Specific Welfare-Focused Remedial Interventions and/or Maintenance Activities

As an extension of Application 3, repeated assessments using the Model potentially enable monitoring of change in all *identifiable* welfare-significant attributes, whether they are negative or positive (e.g., [37]). Thus, deterioration, improvement or stability of these specific attributes may be followed over time and the effectiveness or otherwise of the related management interventions may be evaluated. Note, however, that such monitoring and management can be applied only to those attributes that can be identified specifically under the circumstances in which the monitoring takes place. For example, remote visual observation of appearance, demeanour and behaviour may limit, and detailed hands-on physical/physiological/clinical monitoring may extend, the range of welfare-relevant attributes that can be assessed.

### 4.5. Application 5: Model Use Facilitates Qualitative Grading of Specific Features of Welfare Compromise and/or Enhancement

Animal welfare monitoring (Application 4) requires a capacity to grade the attributes of interest. Accordingly, grading systems have been incorporated into the Model from its original formulation [1,10,11,12,13,37]. The bases for grading welfare compromise and enhancement differ as the defining point of reference for compromise is “suffering” and its mitigation, and for enhancement the focus is on animals’ utilisation of opportunities to experience “positive affective engagement” [12,20]. Thus, the corresponding welfare impact scales also differ, as will now be described.

#### 4.5.1. Grading Welfare Compromise

Compromise may be graded using a five-tier scale (A to E), where grades A and B represent no and tolerably low intensity negative affects, respectively, grade E represents exceptionally unpleasant experiences manifested as negative affects at very high intensities, and grades C and D represent intermediate levels (Table 1). These grades therefore equate to different degrees of welfare compromise ranging from none to very severe [11]. Examples of full Model-based grading of multiple negative affects aligned with Domains 1 to 4 in different contexts have been published elsewhere [1,11,26,27,31,36,37,43], and examples restricted to single Domains are provided here (Table 1).

Grades for compromise related to specific affects are distinguished largely according to the following three criteria: (1) the severity of the physical/functional impacts and of unpleasant external circumstances in Domains 1 to 4; (2) the related intensity and duration of the inferred affective impacts and their reversibility; and (3) whether or not these impacts may need to be urgently mitigated and/or ended by relocation to more benign conditions, by animal care or veterinary therapeutic interventions, and/or by euthanasia [11].

Note that simply because a five-tier scale is notionally available, this does not necessarily mean that grading different features of welfare compromise can be achieved with the degree of precision implied by that number of tiers (e.g., Table 1). For example, when information is sparse or contradictory it may only be possible to distinguish between “no to low”, “moderate” and “severe” compromise, or, at its simplest, when a particular form of compromise is either “absent” or “present” [12,31,32,36]. Note also that numerical grading was explicitly rejected to avoid facile, non-reflective averaging of “scores” as a substitute for considered judgment and to avoid implying, unrealistically, that much greater precision is achievable than is possible with such qualitative assessments [1,13].

#### 4.5.2. Grading Enhanced Welfare

Enhancement is graded using a four-tier scale (0, +, ++, +++), modified from that developed by Edgar and colleagues [87], where the tiers represent “no”, “low-level”, “medium-level” and “high-level” enhancement, respectively (Table 2) [12]. The conceptual framework underpinning application of this scale has three key elements. The first is the availability of *opportunities* for animals to engage in self-motivated rewarding behaviours, and the second is the animals’ actual *utilisation* of those opportunities. Grading *opportunity* and *use* separately helpfully provides more detail to underpin the third element; i.e., the making of cautious judgements about degrees of “positive affective engagement”. Once graded, such positive engagement is taken to be equivalent to the graded extent of *welfare enhancement* for each of the affects inferred to be experienced [20]. Examples of grading are provided elsewhere [12], as also in Table 2.

Thus, three interacting scales representing *opportunity*, *utilisation* and *enhancement* are employed where the tiers for each are “0”, “+”, “++” and “+++”. A full explanation of potential interactions between the three scales is provided elsewhere (Mellor and Beausoleil 2015), as are some examples [12,37]. Key points to note here are as follows:
Opportunity constrains use and therefore use cannot be graded higher than opportunity;For all opportunity grades above zero, use may be graded as equal to or less than opportunity;Use constrains welfare enhancement so that an absence of use precludes enhancement related to the opportunity not utilised;Each use grade above zero, once cautiously interpreted in terms of the possible extent of “positive affective engagement”, is assigned an equal grade on the welfare enhancement scale;Grading the extent of “positive affective engagement” is based on cautious inferences regarding observed behaviours that indicate animals exercising agency in particular ways. Expressed in general terms, the associated positive affects are likely to include various forms of comfort, pleasure, interest, confidence and a sense of control.

Importantly, behavioural-neuroscience evidence suggests that such agency-expressive behaviours (Figure 1) are likely to be affectively rewarding (e.g., [21,22,45]), a conclusion supported by behavioural science observation of animals’ preferences, aversions and priorities (e.g., [45,91,92]).

Rewarding behaviours may arise when the key attributes of animals’ environments include, but are not limited to, the following (Figure 1) (Table 2) [5,12,46]:
Variability that provides a congenial balance between predictability and unpredictability;Access to preferred sites for resting, thermal comfort and voiding excrement;Environmental choices that encourage exploratory and food acquisition behaviours which are enjoyable;Availability of a variety of feeds having pleasurable smells, tastes and textures; andCircumstances that enable social species to engage in bonding and bond affirming activities and, as appropriate, other affiliative interactions such as maternal, paternal or group care of young, play behaviour and sexual activity.

### 4.6. Application 6: Model Use Facilitates Both Prospective and Retrospective Animal Welfare Assessments

The Model has been used prospectively to assess bad and/or good welfare impacts of proposed new or modified approaches to managing, housing and/or interacting with farm [11,33], working [37], zoo [11,30], “pest” [26,27,28,31,32,35,36], research [10,29] and other animals [11]. Moreover, in line with the Model’s original purpose [1], its prospective use to assess potential negative impacts of research, teaching and testing manipulations has been a mandatory part of New Zealand’s code of ethical conduct and animal ethics committee system for regulating animal-based science since 1997 [13].

The Model is also well suited to retrospective welfare analyses to help focus attention on areas of concern and to guide the implementation of remedies [12,31,32,35,36,37], including ways of promoting positive welfare states [18,20,21,22,87]. Retrospective evaluation of predicted negative welfare outcomes of particular activities, for example, research, teaching and testing manipulation of sentient animals, also helpfully provides a check on such predictions and, with or without modification, adds weight to such prospective assessments in future [1,13]. Canadian experience shows that such retrospective assessments using the Model can also enhance the preparation of expert opinions related to prosecutions for serious welfare offences by highlighting scientifically supported connections between indicative physical/functional states and behaviours and their aligned negative affective experiences in ill-treated animals [93].

Finally, combining retrospective and prospective analyses to assess an animal’s current and likely future welfare status, i.e., its quality of life [7,9,12,25], could facilitate end-of-life decisions in a wide range of species and circumstances, because such a use of the Model makes available helpful information for inclusion in the decision-making process (see Application 7).

### 4.7. Application 7: Model Use Facilitates Consideration of Quality of Life (QoL)

By incorporating both negative and positive welfare-relevant experiences, the Model has clear application to consideration of QoL, which, conceptually at least, refers to the net negative-positive affective balance an animal may experience *over an extended period of time* [9,25,37,68,85,87,94,95]. Although the Model can be used to grade both welfare compromise and enhancement, represented as a combined symbol for the factors identified, as briefly explained above (Application 5) and illustrated in more detail elsewhere [12,37], that combined symbol is not an all-inclusive metric for QoL. There are six interacting reasons for this [12,25,32,68,88,94,95]:
The grading of welfare compromise or enhancement is limited to the specific affects inferred to be elicited by the particular internal states or external circumstances that can be identified practically using the indices that are available to be applied in each case, which potentially leaves unassessed welfare relevant factors that cannot be identified in those cases;The relative impacts of each *negative* affect, of each *positive* affect, and between specific *negative* and *positive* affects, are not known; nor are the relative impacts of each such affect within individuals over time or between individual animals;Also unknown are the impacts on the animals’ current and future perceptions of their prior affective experiences, which might have been negative, positive, short-lived, protracted, and/or of low or high intensity;The conceptual foundations for the grading of key elements of compromised and enhanced welfare differ, and are therefore not strictly comparable;The grading of each affect is ordinal, i.e., qualitative, which precludes quantitative comparisons;Each welfare assessment using the Model relates to the animal’s state at the time, and although repeated assessments can provide information about changes in specific elements of welfare over the period observed (e.g., [37]), the other impediments to an all-inclusive QoL assessment remain.

Nevertheless, recall that when the intensity of one or more of the *survival-critical negative affects* is significant, animals are often demotivated from utilising existing opportunities to engage in behaviours that would be accompanied by *situation-related positive affects* (Figure 2) [5,12,37]. This could be informative with regard to end-of-life decision-making. For example, such demotivation may occur in cases where specific areas of significant welfare compromise and their aligned negative affects have been identified, and where there is evidence that the animal rarely, if ever, engages in rewarding behaviours it had enjoyed previously. Clearly, this would be helpful adjunct information about key elements of the animal’s QoL that might support a decision to euthanase it. Conversely, evidence that an animal spontaneously or with encouragement regularly engages in behaviours it obviously finds pleasurable might support a decision not to euthanase it [37].

## 5. Conclusions

The utility of the Five Domains Model for animal welfare assessment is based on validated scientific foundations of the physical/functional and behavioural indices of negative affects aligned with welfare compromise and positive affects aligned with welfare enhancement. The wide range of affects now identified for consideration and the configuration of the domains that was designed specifically to clarify the likely sources of those affects, together enable Model-based welfare assessments to be structured, systematic, comprehensive and coherent. Seven interacting applications of the Model have been described as having the following beneficial objectives—they specify key general foci for animal welfare management; highlight the foundations of specific welfare management objectives; enable monitoring of responses to specific welfare-focused remedial interventions and/or maintenance activities; identify previously unrecognised features of poor and good welfare; facilitate qualitative grading of specific features of welfare compromise and/or enhancement; enable both prospective and retrospective animal welfare assessments to be conducted; and provide adjunct information to support consideration of QoL evaluations in the context of end-of-life decisions. Nevertheless, it is important not to overstate what utilisation of the Model can achieve. Constraints arise through the following factors: different levels of confidence with which particular affects may be inferred to be present in different circumstances; the necessary focus only on the specific affects that can be identified; differing precision with which each affect may be graded; and the limits imposed by an inability to determine the relative impacts of different affects when evaluating the notional overall negative-positive affective balance represented by QoL, thereby precluding the possibility of elaborating an all-inclusive QoL metric.

## Figures and Tables

**Figure 1 animals-07-00060-f001:**
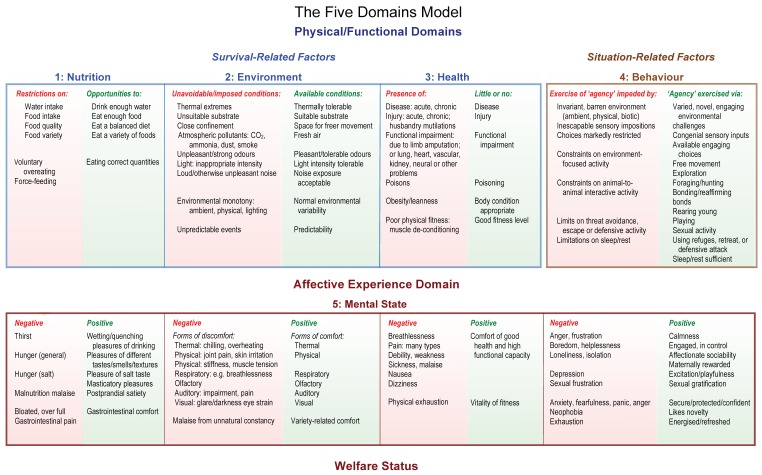
The Five Domains Model (modified from [12]): The examples provided for the physical/functional Domains 1 to 3, labelled “Nutrition”, “Environment” and “Health”, are intended to direct attention towards mainly *internal* survival-related factors, and those provided for Domain 4, labelled “Behaviour”, focus attention largely on *external* situation-related factors. For each of Domains 1 to 4, examples of negative and positive factors are provided and are aligned with inferred negative or positive affective experiences, assigned to Domain 5, labelled “Mental State”. The overall affective experience in the mental domain equates to the welfare status of the animals, as explained in the text. Note that an animal exercises “agency” (Domain 4: “Behaviour”) when it engages in voluntary, self-generated and goal-directed behaviours [44,45].

**Figure 2 animals-07-00060-f002:**
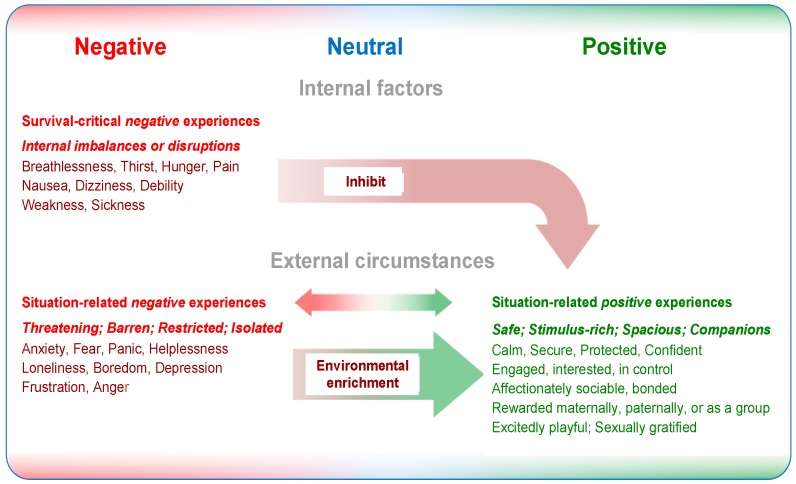
Depiction of different subjective experiences, or affects, over the full valence range from negative-to-neutral-to-positive and relationships between the different types of experience. **Internal factors**, which include naturally occurring or induced functional imbalances or disruptions (captured mainly by Domains 1 to 3), give rise to *survival-critical experiences* (e.g., breathlessness, thirst, hunger, pain, nausea, sickness) that motivate animals to engage in behaviours aimed at securing life-sustaining resources (e.g., oxygen, water, food) or minimising life-threatening harms (e.g., injury, food poisoning, infection). The valence of these experiences is negative, and their intensity ranges from exceptionally negative to neutral. **External factors**, which influence animals’ perception of the levels of threat or safety, degrees of under-stimulation or pleasurable stimulation from low to high, restrictions on or ease of movement, and social isolation or opportunities for companionable interaction with other animals (captured mainly by Domain 4), give rise to *situation-related experiences* over the full valence range from strongly negative to strongly positive. *Environmental enrichment* initiatives can replace situation-related negative experiences with positive experiences. **Interactions** between the different types of experience are apparent when the intensity of *negative survival-critical experiences* is sufficiently severe to demotivate or *inhibit* animals from utilising available opportunities to engage in behaviours that would generate *positive situation-related experiences*.

**Table 1 animals-07-00060-t001:** Model-based grading of animal welfare compromise related to particular challenges. The examples refer to specific indices and inferred affects considered in relation to Domains 1 to 4, except for toxicity testing in Domain 3 where the observed indices and inferred affects are generalised. Example animals include livestock, working animals, pets, “pests” and laboratory animals. Note that, theoretically, Model-based grading may be applied to any vertebrate where scientific understanding is sufficient to support the meaningful use of particular indices. Note also that the primary purpose here is to illustrate how specific attributes may be graded. It is not to demonstrate a full Model-based assessment of compromise involving grading of the multiple attributes covered via Domains 1 to 4 and their inferred affects via Domain 5, all considered together (Figure 1). Details of such full assessments have been published elsewhere [26,27,28,35,36,37]. Finally, note that “A: none” on the compromise scale does not imply welfare enhancement (see Table 2).

Animal Welfare Challenge	Compromise Grade				
	A: None	B: Low	C: Mild to Moderate	D: Marked to Severe	E: Very Severe
*Domain 1: Nutrition*					
*Access to water* in livestock, pets, working animals, etc.:	Water freely available:	12-h interruption in water supply, cold weather:	24-h interruption in water supply; hot weather:	Within-group competition for limited water long term:	Water not available (supply failure, drought):
Availability; inferred thirst	No to very low-level thirst	Low-level thirst	Moderate thirst	Severe thirst	Extreme thirst
*Feeding level* in sheep:	Good-level and stable body condition (3/5):	Mid-level and stable body condition (2.5/5):	Mid-level body condition (2.5/5), slowly decreasing:	Rapidly decreasing or low-level body condition (1.5/5):	Very low body condition (0.5/5)—emaciated:
Body condition score; inferred hunger	No to very low-level hunger	Low-level hunger	Moderate hunger	Severe hunger	Extreme hunger
*Domain 2: Environment*					
*Heat load* in sheep: Panting; inferred hyperthermic distress	Ambient conditions thermoneutral:		High radiant load, temperature, humidity:		Extreme radiant load, temperature, humidity:
	No panting		Closed mouth panting		Open mouth panting
	No hyperthermic distress		Mild to moderate distress		Very severe distress
*Air quality* in housed pigs: NH_4_ levels; inferred eye and nasal irritation	Good ventilation, fresh air: No eye/nasal irritation		Ventilation poor:	Ventilation very poor:	
			NH_4_ 10–15 ppm	NH_4_ greater than 25 ppm	
			Mild eye/nasal irritation	Marked eye/nasal irritation	
*Domain 3: Health*					
*Amputation dehorning* in calves:	Nerve blockade plus systemic analgesic:		Nerve blockade alone or systemic analgesic alone:		No pain relief:
Acute cortisol stress response; inferred pain	Complete pain relief		Partial pain relief		
	Very low stress response		Moderate to marked stress response		Very marked stress response
	Little or no acute pain		Moderate to marked acute pain		Very marked acute pain
*Impeded breathing* in dogs: Exercise intolerance; inferred breathlessness	Normal or long-nosed:		Moderately snub-nosed:		Severely snub-nosed:
	Exercise tolerant, breathing normal		Brief exercise bouts ended by laboured breathing		Laboured breathing at rest, totally exercise intolerant
	No breathlessness		Moderate breathlessness		Very severe breathlessness
*Toxicity testing* in pest and laboratory animals:	Non-toxic substances:	Low toxicity substances:	Mildly toxic substances:	Markedly toxic substances:	Highly toxic substances:
Untoward organ-specific clinical signs; various affects	No untoward clinical signs	Minor/short lived clinical signs, then recovery	Moderate/short lived or minor/longer lived clinical signs, then recovery	Marked/short lived or moderate/longer lived clinical signs, then recovery	Extreme clinical signs, followed by death while conscious
*Domain 4: Behaviour*					
*Tethering/caging of* dogs:	Not tethered/caged:	Tethered/caged 25% of the time:	Tethered/caged 50% of the time:	Tethered/caged 75% of the time:	Tethered/caged 100% of the time:
Exercise limitation; inferred boredom/depression	Exercise not limited	Some boredom/depression	Medium boredom/depression	Marked boredom/depression	Severe boredom/depression
	No boredom/depression				
*Handling* livestock:	Calm, tamed, trained and fully compliant animals:	Feedlot animals with regular human contact:	Paddock animals with some human contact:	Range animals with little prior human contact:	Feral/wild animals with no prior human contact:
Prior contact; restraint level; induced cortisol stress response; inferred fear	Gentle handling	Need light restraint	Need firm restraint	Need strong restraint	Need very strong restraint
	No response and fear	Low response and fear	Moderate response and fear	Marked response and fear	Extreme response and fear

Relevant major sources: [5,6,12,16,17,18,20,21,22,29,34,44,46,56,57,58,59,61,66,67,68,69,70,71,72,73,74,75,76,77,78,79,80,81,82,83,84,85,86,87,88,89,90].

**Table 2 animals-07-00060-t002:** Grading of animal welfare enhancement via opportunities to engage in an increasing range of rewarding behaviours in each of Domains 1 to 4. The grades relate to the observed utilisation of available opportunities to experience various forms of comfort, pleasure, interest, confidence and a sense of being in control, so that the primary indices are behavioural (Domain 4). Example animals are livestock (Domain 1), pigs and laboratory animals (Domain 2), and a wide range of terrestrial mammals (Domains 3 and 4). The primary purpose here is to illustrate how various combinations of Domain-related specific opportunities may be graded, not to demonstrate a full Model-based assessment of enhancement involving the multiple opportunities covered via Domains 1 to 4 and their inferred affects via Domain 5, all considered together (Figure 1). Details of such full assessments related mainly to mammals have been published elsewhere [12,37]. Note that key features of a three-tier system for welfare enhancement in poultry [87] could provide a basis for modifying the Five Domains Model to more directly include avian species in its applications.

Domain	Animal Welfare Enhancement Opportunities			
	None (o)	Low-Level (+)	Mid-Level (++)	High-Level (+++)
*Domain 1: Nutrition*	Quantity and quality meet functional needs	Quantity and quality meet functional needs	Quantity and quality meet functional needs	Quantity and quality meet functional needs
Livestock fed indoors and/or outdoors	Diet components and palatability constant over long periods	Very limited choice among diets with pleasant smells, tastes and textures via food supplements or outdoor seasonal changes	Moderate choice among varied diets with pleasant smells, tastes and textures available indoors and/or outdoors	Widely varied diets enabling choices between pleasant food smells, tastes and textures in engagingly different locations
*Domain 2: Environment*	Monotonous ambient, physical and lighting conditions	Marginal increase in space allows freer movement	Moderate increase in space allows greater separation between resting animals	Space sufficient for separate eating, resting and dunging sites
Groups of pigs kept indoors	Limited space restricts animals’ activities	Deep, clean, dry floor substrate	Deep, clean, dry floor substrate	Space for calm social interaction
Groups of laboratory animals kept indoors	Bare floor		Refuges	Deep, clean, dry floor substrate; Refuges Air temperature variations aid comfortable thermoregulation
*Domain 3: Health*			Good health actively maintained:	Good health actively maintained:
A wide range of mammals:			Disinhibited from engaging in rewarding behaviours	Disinhibited from engaging in rewarding behaviours
Health management;				Exercise opportunities help maintain good physical fitness
Degree of physical fitness				
*Domain 4: Behaviour*		Limited opportunities for positively	Greater opportunities for positively	Diverse opportunities for positively
A wide range of mammals:		motivated exploration, food acquisition,	motivated exploration, food acquisition,	motivated exploration, food acquisition,
Rewarding behaviours		bonding, care of young, play or sexual activity	bonding, care of young, play or sexual activity	bonding, care of young, play or sexual activity

Relevant major sources: [1,5,6,9,12,14,15,16,20,21,22,32,36,41,42,46,48,53,54,66,67,68,69,70,71,72,73,74,75,76,77,78,79,80,81,82,83,84,85,86].

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
