# Peer review of "Operational Details of the Five Domains Model and Its Key Applications to the Assessment and Management of Animal Welfare"

_animals, 2017, doi:10.3390/ani7080060_

Round 1

Reviewer 1 Report

This paper is a valuable addition to the literature the author has developed on the Five Domains Model and its uses, and the author should be commended. Comments made below are of a minor nature, some for clarification of meaning and some minor errors.

Comments

I don’t agree with the depiction in Figure 2, whereby the survival-critical experiences are only given a negative to neutral range. For example, in the Nutrition Domain, eating sufficient and a variety of foods may support a positive welfare state to the animal, rather than merely being neutral?

Lines 252-54: The author states there is underlying neuroscience knowledge, physical/functional indices and behaviours to allow pain, amongst the other affects, to be identified.  The statement infers an identification, i.e. pain is present or it is not present, and I would counter that our measures of pain are not reliable enough to be able to do this at all times. Pain is a subjective experience of an individual animal and difficult to assess with accuracy. In this paragraph sickness is listed as something that can be identified, but then further down as one of the affects that it is not easy to distinguish between. The wording needs to be clarified.

Lines 261-269: Although there are good neuroscience bases for distinguishing between anxiety, fear, panic etc, these are experimental and not validated in all species of animal. In a practical sense I think it is more difficult to distinguish between these affects.

Line 295: Even though hearing in the ultrasonic range may not be something humans can do, there is credible scientific evidence that loud noises in the ultrasonic range will negatively affect welfare, for eg. in laboratory animal facilities rodents may no longer breed if these noises occur due to building works. In addition, rats emit sounds in the ultrasonic range when they are tickled, aligned with this as a positive experience (Panksepp).

Line 386: for animals to have positive…

Author Response

I am happy for the referee to see the entire document I have uploaded.

Reviewer 2 Report

Operational details of the five domains model and its key applications to the assessment and management of animal welfare

The five domains model was originally proposed by the present author in the 1990s. Since then it has been used in a wide variety of welfare applications, and the author and colleagues have also published various refinements to the model which have increased its utility and reliability.

This review makes an extremely useful summary of this work and of the further potential utility of the model. However, I found the seven sections on potential applications a bit dry and hard to follow/distinguish and I think these would benefit enormously by basing each one around a real or hypothetical example of the particular application concerned. I think this would make the meat of the paper more readable.

Apart from incorporating these examples, I have the following small suggestions:

Line 188 - 'voluntarily' should read 'voluntary'.

Line 201 - suggest adding ‘greater’ to read “i.e. they have greater opportunities…”

Figure 2 - suggest adding to the figure (and to the figure legend) which domains relate to which experiences, i.e. Domains 1-3 with survival-critical and Domain 4 with situation-related’.

Line 338 - clarify in the heading that this relates to captive situations.

Line 386 – add ‘to’ to read “for animals to have…”

Line 392-394 – please give example.

Line 404-406 – example please.

Line 435-436 – this last sentence seems a bit of an abrupt way to end the section. Could it be incorporated earlier or otherwise reworded?

Lines 520-524 - Baker, Sharp and Macdonald (2016) also used an adaptation of the model for modelling the welfare impact of a wildlife management method and then also for identifying ways to refine methods to improve welfare. You could cite this here too?

Lines 534-536 - Baker et al (mentioned above) highlight similarities between Sharp and Saunders’ adaptation of the 5 domains model and the QALYs system used in the allocation of healthcare resources in the UK. See http://www.vhpharmsci.com/decisionmaking/Therapeutic_Decision_Making/Advanced_files/What%20is%20a%20QALY.pdf. Might be worth mentioning this?

Line 554-555 – more explanation needed as to why these relative impacts aren’t known. Surely you can compare two scores for negative welfare impact and see if one is worse than another? Similarly you can compare a negative and a positive score and say that the positive score is better than the negative one? Or are you saying that where the types of impacts are different the scales are not comparable?

Author Response

(The authors gave the same response as above.)

Reviewer 3 Report

This is a clear description of the developing concepts and applications of the "Five Domains" model.  However, I believe that it could be improved by less reiteration of the principles of the model itself and the inclusion of at least two examples of the application of the model to welfare assessment and management in practice (e.g. on a farm and in a research laboratory).

In App 3 (l403) suffering is described as an imprecise term.  The Broom/Webster definitions of suffering are quantitatively quite precise; defining the point beyond which an animal fails to cope adequately with challenges that are too severe, too complex or too prolonged.

Author Response

I am happy for the referee to see the entire document I have uploaded
